# A Sentinel-2 Image-Based Irrigation Advisory Service: Cases for Tea Plantations

Yi-Ping Wang [1], Chien-Teh Chen [2], Yao-Chuan Tsai [3] and Yuan Shen [1,4,*]

[1] Department of Soil and Environmental Sciences, College of Agriculture and Natural Resources, National Chung-Hsing University, Taichung City 402, Taiwan; shuziwang@yahoo.com.tw

[2] Department of Agronomy, College of Agriculture and Natural Resources, National Chung-Hsing University, Taichung City 402, Taiwan; ctchen41@dragon.nchu.edu.tw

[3] Department of Bio-Industrial Mechatronics Engineering, College of Agriculture and Natural Resources, National Chung-Hsing University, Taichung City 402, Taiwan; yctsaii@dragon.nchu.edu.tw

[4] Innovation and Development Center of Sustainable Agriculture, National Chung-Hsing University, Taichung City 402, Taiwan

[*] Correspondence: yshen@nchu.edu.tw; Tel.: +886-4-2287-2619; Fax: +886-4-2286-2043

**Abstract:** In this study, we aim to develop an inexpensive site-specific irrigation advisory service for resolving disadvantages related to using immobile soil moisture sensors and to the differences in irrigation needs of different tea plantations affected by variabilities in cultivars, plant ages, soil heterogeneity, and management practices. In the paper, we present methodologies to retrieve two biophysical variables, surface soil water content and canopy water content of tea trees from Sentinel-2 (S2) (European Space Agency, Paris, France) images and consider their association with crop water availability status to be used for making decisions to send an alert level. Precipitation records are used as auxiliary information to assist in determining or modifying the alert level. Once the site-specific alert level for each target plantation is determined, it is sent to the corresponding farmer through text messaging. All the processes that make up the service, from downloading an S2 image from the web to alert level text messaging, are automated and can be completed before 7:30 a.m. the next day after an S2 image was taken. Therefore, the service is operated cyclically, and corresponds to the five-day revisit period of S2, but one day behind the S2 image acquisition date. However, it should be noted that the amount of irrigation water required for each site-specific plantation has not yet been estimated because of the complexities involved. Instead, a single irrigation rate (300 t ha$^{-1}$) per irrigation event is recommended. The service is now available to over 20 tea plantations in the Mingjian Township, the largest tea producing region in Taiwan, free of charge since September 2020. This operational application is expected to save expenditures on buying irrigation water and induce deeper root systems by decreasing the frequency of insufficient irrigation commonly employed by local farmers.

**Keywords:** DSS for irrigation; water use efficiency; soil moisture; Sentinel-2; tea



## 1. Introduction

Taiwan is an island about 400 km long and 150 km wide, located at the southeast edge of the Asia continent. About two-thirds of the island is mountains and a series of mountain ranges, formed by over 260 peaks higher than 3000 m, bisects the island from north to south. Uneven rainfall distribution among seasons, short but rapid flow rivers, and sedimentations at main reservoirs make Taiwan a water-starved island, although it receives an average annual rainfall of 2500 mm [1]. Water use efficiency and saving are the keys to balance competitions between agricultural and industrial sectors and contribute to social justice [2]. Currently, there are still no irrigation advisory services in Taiwan to help local farmers to increase water use efficiency, particularly for upland crops. The impact of climate change on the agricultural sector due to changing temperature and rainfall patterns

further highlights the need for modern technologies to improve water use efficiency so that the already scarce freshwater resource can be applied more precisely and effectively [3].

Tea is an important economic crop in Taiwan. However, the guidance for irrigation practice from the Tea Research and Experimental Station (TRES), a governmental professional organization for guidance and promotion of the Taiwan tea industry, is only a single instruction, i.e., irrigation duration of 3 h, at intervals of 5–7 days [4], without considering the plant, soil, and weather conditions. Therefore, farmers generally practice irrigation based on their own experiences. On average, irrigation charges of an established and producing tea plantation make up about 14% of the total production costs, next to labor and fertilizer costs [5]. Therefore, developing a site-specific irrigation advisory service to increase water use efficiency of tea plantations would be useful not only to meet Sustainable Development Goal (SDG) 6.4 of the United Nations [6] but also to increase farmers' income. However, due to the small-scale farming system adopted in Taiwan, the average size of a tea plantation is about 0.3 ha. Therefore, landscapes are very fragmented because of variations in tea tree cultivars, plant ages, and management practices adopted by each farmer. This makes the development of an irrigation advisory service offering site-specific recommendations even more difficult.

Soil moisture sensors are proximal instruments commonly used to control the timing of watering [7,8]. The "start/stop" irrigation control is generally based on readings of two or more sensors buried underneath the root zone of plants. Recent advances in sensor technology and the evolution of wireless sensor network (WSN) and internet of things (IoT) technologies have now been widely applied, together with soil moisture sensors, to develop smart irrigation systems for agriculture [9,10]. However, the approach using soil moisture sensors has been argued to have important disadvantages due to soil heterogeneity and the fact that crop water status depends on soil moisture content, as well as other abiotic and biotic factors [11–15]. In addition, the initial investment in wireless soil moisture sensors and the cost for instrument maintenance are prohibitively high for farmers.

Using agrometeorological data from a nearby weather station is another popular approach for estimating crop water requirements [16,17]. Other than computing reference evapotranspiration (ETo) from meteorological data, a crop coefficient (Kc) that represents the physical and physiological differences between the target crop and the reference crop must be provided to estimate actual evapotranspiration loss from a field [16]. This requirement can severely limit this approach's application for an intended site-specific service due to the difficulty in assigning proper coefficient values for each plantation individually.

In addition to approaches using in situ measurements, remote sensing is increasingly used to provide information on irrigation requirements assessment at multiple spatial and temporal scales. The Thermal-Optical Trapezoid Model (TOTRAM), based on the remotely sensed land surface temperature (LST) and vegetation index, is one of the most widely applied approaches for estimating surface soil moisture or actual evapotranspiration [18]. However, the application of TOTRAM suffers from two inherent limitations. It requires simultaneous optical and thermal observations and individual parameterization/calibration for the LST band of each observation date [19].

Sentinel-2A/-2B (S2), launched by the European Space Agency, as part of the Copernicus program [20], provide free and open access high spatiotemporal resolution satellite imagery. The five-day S2 acquisition frequency is adequate to create sufficiently dense time series for agronomic monitoring purposes. Meanwhile, its decametric spatial resolution allows discriminating the spatial variability at inter- or intra-field scales. Although an S2 image has 13 spectral bands in the optical domain, it has no thermal band. This excludes the approach using the TOTRAM model. However, Sadeghi et al. [21] proposed an Optical TRApezoid Model (OPTRAM) based on the linear physical relationship between soil water content (SWC) and shortwave infrared (SWIR) transformed reflectance (STR), to estimate surface soil moisture using Sentinel-2 images. They also showed that the OPTRAM

model required only a single universal parameterization for a given location, which was a significant advancement for remote sensing of soil moisture.

Using laboratory measurements and simulations by a leaf radiative transfer model, Ceccato et al. [22] showed that SWIR was sensitive to vegetation water content; however, it could not be used alone to retrieve vegetation water content because two other leaf parameters (internal structure and dry matter) also influenced SWIR leaf reflectance. A combination of SWIR and near-infrared (NIR, only influenced by internal structure and dry matter) is necessary to retrieve water content at the leaf level. The normalized difference water index (NDWI) [23], global vegetation moisture index (GVMI) [24], and shortwave infrared water stress index (SIWSI) [25] are vegetation indices (VIs) that use SWIR and NIR band reflectance from remotely sensed images, in a very similar normalized difference form, to retrieve information on short-term variations in canopy water content. Considering atmospheric transmission properties and the incident solar spectral irradiance at the earth's surface, Tucker [26] suggested that the 1.55–1.75 μm spectral interval was the best-suited band for monitoring water status in plant canopy by airborne or spaceborne optical sensors. The central wavelength of Sentinel-2 Band 11 locates in this spectral interval.

The normalized difference vegetation index (NDVI) is the most commonly used VI to correlate with various vegetation biophysical variables such as biomass, leaf area index, fractional vegetation cover, and the fraction of absorbed photosynthetically active radiation [27,28]. However, the NDVI has been shown to be very sensitive to soil optical properties at incomplete vegetation cover conditions [29]. In order to reduce the effect of soil background, several modifications of NDVI have subsequently been proposed, such as the soil-adjusted vegetation index (SAVI) [30], transformed soil adjusted vegetation index (TSAVI) [31], modified soil-adjusted vegetation index (MSAVI) [32], optimized soil adjusted vegetation index (OSAVI) [33], and generalized soil-adjusted vegetation index (GESAVI) [34]; however, except for the MSAVI, all other soil-adjusted VIs use a constant soil adjustment factor in their equations. In addition, Huete [24] found that the optimal soil adjustment factor should be varied with vegetation density. The MSAVI was found to be more flexible for reducing soil background influences and in better agreement with vegetation biophysical variables than other soil-adjusted VIs [35].

As analyzed above, an S2 image-based irrigation advisory service may provide the flexibility for providing site-specific irrigation recommendations at a minimum cost. Therefore, the main goals of this study are:

(1)  to evaluate the possibility of retrieving crop water availability status from tea plantations using S2 satellite data,
(2)  to build an irrigation advisory service based on the retrieved crop water availability status, and
(3)  to evaluate the abilities/potentials of the developed service to support tea farmers in irrigation management.

## 2. Materials and Methods

### 2.1. Study Region

The study region is located at Mingjian Township (23°48′5.184″–23°53′24.324″ N, 120°36′55.044″–120°44′14.136″ E), which is the largest tea-producing region in Taiwan. It is a plateau with altitudes falling gently from 430 to 200 m above sea level. The topography is higher to the southwest border and declines gently towards the eastern border. The acidic soils, which are comprised of Oxisols and Ultisols, are favorable for tea and pineapple production. In addition, ginger, Chinese yam, and betel pepper are also popular crops in the Mingjian Township. The climate is classified as Cwa by the Köppen classification system. The mean monthly temperatures range from 20 °C in February to 32 °C in August and the annual rainfall ranges from 1500 to 2000 mm, with the dry season generally from October to March.

Land use is composed of tea plantation (26.1%), other crops (37.1%), forest (15.6%), built (17.1%), and waterbody (4.1%). The primary tea tree cultivar planted in the region is Si

Ji Chun (Four Season Tea), but cultivars such as Jin Xuan (TRES#12), Ruby (TRES#18), and Qin Xing Oolong can also be found. Each plantation's cultivation management practices have been determined by individual farmers based on their preferences and experiences. Significant differences that may affect the canopy reflectance are weed control, plant height, and open space between rows. Herbicides, dark- to light-colored plastic films or fabric mats, or mulches of peanut shells and rice husks are commonly used to suppress weeds. Plantations of tree height from 1.5 to 0.3 m and percent ground cover from fully covered to about 50% covered are common in the region. Most tea plantations have installed sprinkler irrigation systems using water transmitted through a pipe from a nearby storage pool to avoid drought stress. The drip irrigation system is sporadic in the region. Farmers usually start the irrigation system when the soil surface looks dry.

A total of 121 tea plantations spread across the study region, selected as study sites for a soil health diagnosis project concurrently conducted in the region, were routinely visited for plant health checkups. These plantations were used to determine the thresholds for the classification of surface soil moisture content and canopy water content of tea trees, as explained in more detail later.

### 2.2. Image Acquisition and Processing

Sentinel-2 Level-1C (S2 L1C) images starting from 2018 and at tile T51QTG, in which the study region is located, were acquired from the Copernicus Open Access Hub [36]. The downloaded S2 L1C images were first processed by FMASK (V.4.2 standalone, [37]) to produce a mask for pixels affected by cloud and cloud shadow, and by Sen2Cor (V.2.8, [38]) to convert the L1C product (top-of-atmosphere reflectance) to L2A product (surface reflectance). The cloud mask and converted L2A product were clipped over the study region's extent and automatically kept in the image archive as soon as new images were available and with less than 50% cloud-affected pixels within the study region. The stored cloud masks and L2A products were further resampled to 6 m resolution using the nearest neighbor method to match the pixel resolution of SPOT 6 and 7 multispectral products and better delineate the narrow boundary of tea plantations that may be encountered.

The relationships for the dry and wet edges of the OPTRAM model, proposed by Sadeghi et al. [21] for computing SWC in tea plantations, then, were derived using all available S2 images from 2018 to June 2020 with three modifications:

(1)  the pixel distribution within the STR-NDVI space was replaced by the STR-MSAVI space to minimize soil background influences on canopy spectra [32],
(2)  only tea plantation pixels were used in deriving the STR-MSAVI space to reduce interferences from other land covers, and
(3)  second-order polynomials fitted to those 2nd and 99th percentile values within each 0.05 bin of MSAVI values were used to define the dry and wet edges of the OPTRAM model for a better fit of the pixel distribution within the STR-MSAVI space.

The second-order polynomials used to describe the dry and wet edges for the study region were listed in Equations (1) and (2), respectively, as follows:

$$\text{STR (dry)} = 0.15 - 0.994 \text{ MSAVI} + 4.234 \text{ MSAVI}^2 \tag{1}$$

$$\text{STR (wet)} = 55.61 - 111.64 \text{ MSAVI} + 68.095 \text{ MSAVI}^2 \tag{2}$$

### 2.3. Development of Irrigation Advisory Service

A two-tiered approach was used to develop the irrigation advisory service (Figure 1). Although each S2 image was taken at approximately 10:30 a.m. local time on the scheduled day, it was not downloaded until 4 a.m. the next day because of the processing time required by the European Space Agency (ESA), internet transmission jam during the daytime, and the need to avoid disturbing farmers' rest at night. Therefore, the service was operated cyclically corresponding to the five-day revisit period of Sentinel-2A/-2B, but 1 day behind the S2 image acquisition date. Considering the possibility of S2 images

obscured by clouds and/or rain after the scheduled S2 image was taken, the first tier was a crop water availability index (CWAI) evaluation matrix, which has six levels and is determined jointly by surface soil water content (SWC) and the normalized difference water index (NDWI), two indices that directly relate to soil wetness and canopy water content, respectively. Both were retrieved from each of the three Sentinel-2 images (images taken on current 1 day, current 6 day, and current 11 day) as shown in Table 1. The four SWC and NDWI levels were determined from their frequency distribution at the 121 plantations within the region from 2018 to June 2020 (explained in more detail later in Section 3.2). The severity of drought stress is negatively related to the CWAI value. A value of CWAI ≥ 4 means there should be no drought stress problem but may have an over-irrigation problem instead.

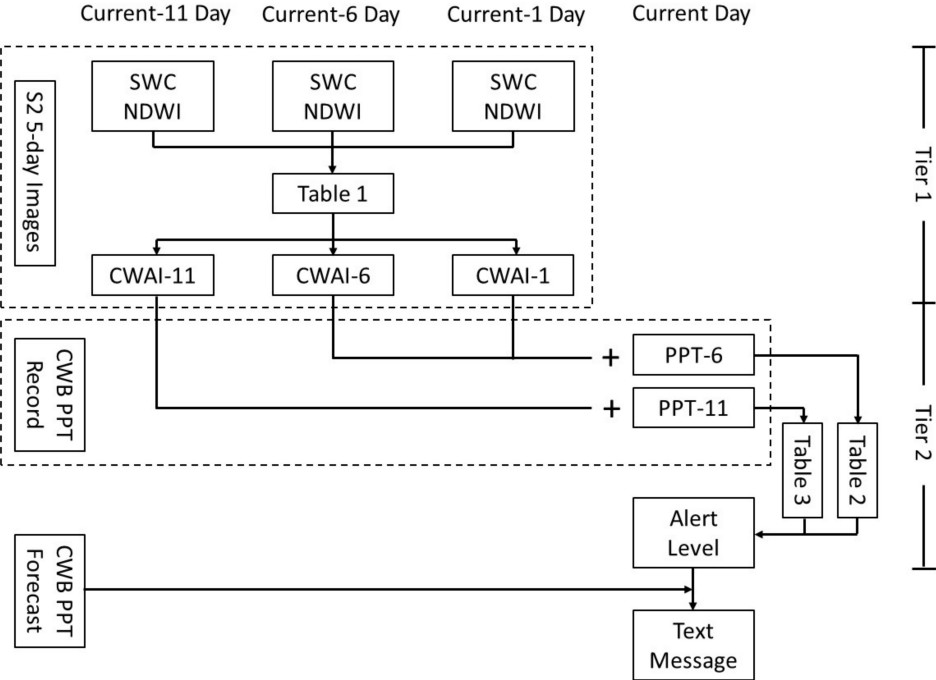

**Figure 1.** Schematics showing the two-tiered approach used in developing the irrigation advisory service. Abbreviations used are explained in the text.

**Table 1.** Evaluation matrix for determination of the crop water availability index (CWAI).

| NDWI | SWC | | | |
|------|------|------|------|------|
| | ≥0.15 | 0.15–0.11 | 0.11–0.08 | <0.08 |
| ≥0.22 | 6 | 5 | 4 | 3 |
| 0.22–0.17 | 5 | 4 | 4 | 3 |
| 0.17–0.11 | 4 | 4 | 3 | 2 |
| <0.11 | 3 | 3 | 2 | 1 |

The SWC was calculated using the second-order polynomials, derived in Section 2.2, for the optical trapezoid's dry and wet edges, i.e., Equations (1) and (2), respectively. The values for STR [21], MSAVI [32], and NDWI [23] of the pixel were calculated based on Equations (3)–(5), respectively, as follows:

$$STR = \frac{(1 - B12)^2}{2} \tag{3}$$

$$MSAVI = \frac{2B8 + 1 - \sqrt{(2B8 + 1)^2 - 8(B8 - B4)}}{2} \tag{4}$$

$$\text{NDWI} = \frac{(\text{B8} - \text{B11})}{(\text{B8} + \text{B11})} \tag{5}$$

where B4, B8, B11, and B12 are S2 bands centered around 665, 833, 1612, and 2202 nm, respectively.

The second tier determines the alert level based on the derived CWAI from the past three S2 images and hourly precipitation record published by the Central Weather Bureau (CWB) [39], as shown in Tables 2 and 3. Once the alert level was determined, it was sent to collaborating farmers through text messages together with precipitation forecasts for the next three days published by CWB [40]. The necessity to start irrigation is also negatively related to the alert level. If classified at alert level 1, the situation is critical, and better to start irrigation soon. No need to irrigate is recommended for the other two alert levels. Since a simple "on/off" switch was the "start/stop" control for the irrigation systems in most tea plantations, the average of the CWAI values of pixels within the target plantation was used for determining the alert level in this study. However, a CWAI map, following the same protocols described below, could be provided if a variable rate irrigation system was installed or maps showing the uniformity of water spraying were needed.

If the plantation was not obscured by clouds in the current-1 day S2 image, the alert level was mainly determined by the plantation's average CWAI from the current-1 day S2 image (CWAI-1 in Table 2). The average CWAI value from the current-6 day S2 image, if available (CWAI-6 in Table 2), was used to smooth the potential minor fluctuations in CWAI values resulting from variations in atmospheric transmission characteristics and instrument noises in L1C product. If the alert level was 1 as determined from the current-1 day S2 image but has $\geq$10 mm rainfall after the image was taken, the alert level is adjusted to level 2 to save irrigation water. Suppose the plantation is obscured by clouds in the current-1 day S2 image, then, the alert level is determined by the average CWAI of the plantation from the current-6 day S2 image and accumulated rainfall from the current day (CWAI-6 and PPT-6 in Table 2). If the value of PPT-6 is < 10 mm and the value of CWAI-6 is <4, the alert level is set to 1 to make sure the tea trees are not severely affected by drought stress.

Suppose the plantation is obscured by clouds in both the current-1 day and current-6 day S2 images and accumulated rainfall from current day is <10 mm, then, the alert level is determined by the average CWAI of the plantation from the current-11 day S2 image and accumulated rainfall from the current-11 day (CWAI-11 and PPT-11 in Table 3). If the value of PPT-11 is <20 mm and the value of CWAI-11 is <4, the alert level is set to 1 to make sure the tea trees are not severely affected by drought stress. Suppose the plantation is obscured by clouds in all three S2 images (current-1 day, current-6 day, and current-11 day) and the value of PPT-11 is <20 mm, the alert level is set to X, representing that the alert level cannot be determined due to insufficient information. Although the same scheme can be applied to use S2 image(s) on the current-17 day (and beyond) to reduce the chance of setting the alert level to X, the chance of making unsuitable recommendations is also increased.

The average daily evapotranspiration in the study region ranges from about 1 mm day$^{-1}$ in January to 3 mm day$^{-1}$ in June and July [41,42]. Therefore, the minimum rainfall thresholds were set at 10 and 20 mm in Tables 2 and 3, respectively, because the revisit period of Sentinel-2A/-2B is 5 days.

All the processes for the advisory service, including S2 image acquisition and preprocessing, CWB webpage data downloading and processing, retrieval of the SWC and NDWI from preprocessed images, alert level determination, and text message sending, were automated and could be completed within 3 h under the computing environment used (Intel Core i7-8700 CPU @3.20 GHz, 16 GB RAM, 64 bits Window 10 OS). The irrigation recommendation was sent before 7:30 a.m., within 24 h after the image was taken, and thus could be considered to be a near real-time service. The advisory service was officially operated from September 2020 and is now serving over 20 collaborating farmers free of charge.

**Table 2.** Decision matrix for alert level determination for cases where values of CWAI-1 and/or CWAI-6 are available.

| CWAI-6 | CWAI-1 | | | | |
|---|---|---|---|---|---|
| | ≥4 | 4–3 | <3 | NA, PPT-6 [2] (mm) | |
| | | | | ≥10 | <10 |
| ≥4 | 3 | 3 | 2 | 3 | 2 |
| 4–3 | 3 | 2 | 1 [1] | 2 | 1 |
| <3 | 3 | 2 | 1 [1] | 2 | 1 |
| NA | 3 | 2 | 1 [1] | 2 | Table 3 |

[1] Adjust to 2 if there was ≥10 mm rainfall after the current-1 day S2 image was taken. [2] Accumulated rainfall from current-6 day to current day.

**Table 3.** Decision matrix for alert level determination for cases where both values of CWAI-1 and CWAI-6 are not available.

| CWAI-11 | PPT-11 [1] (mm) | |
|---|---|---|
| | ≥20 | <20 |
| ≥4 | 3 | 2 |
| 4–3 | 2 | 1 |
| <3 | 2 | 1 |
| NA | 2 | X |

[1] Accumulated rainfall from current-11 day to current day.

*2.4. Ground Data Collection*

Wireless volumetric soil water content sensors (FDR-E018 Rockabye System, Hugreen, Taipei, Taiwan) were installed in five nearby plantations, owned by three farmers, to measure the variations of SWC in the fields. The sensors were installed at 15 and 30 cm depth with at least four replicates at each depth. Only one plantation had continuous SWC data from March to August 2020. The other plantations had SWC data for about 3 months because the number of sensors was limited and they were rotated between plantations. Yearly irrigation charges from 2017 to 2019 of these plantations were also collected and converted to the amount of irrigation water used per unit area to compare these three farmers' irrigation habits.

The uniformity of water delivered by the sprinkler system in each plantation was also evaluated by placing huge funnels, 25 cm in diameter, 8 cm in sidewall height, and with a rubber stopper at the outlet end to prevent leak, at a grid spacing of 10 × 10 m throughout the entire plantation. The sprinkler system was operated by farmers using the same group setup of water filling pipes (as explained in Section 3.4) but run for only 10 min for each group. Then, the spatial distribution of water volumes collected in the funnels was plotted using geostatistics software GS+ (V5, Gamma Design Software LLC, Plainwell, MI, USA).

During the study period, weather information (air temperature, humidity, solar radiation, wind speed and direction, and rainfall) was collected by a weather station (6152C Vantage Pro2, Davis Instrument, Hayward, CA, USA) located less than 1 km from the plantations with SWC measurements.

**3. Results and Discussions**

*3.1. Changes in Soil Water Content (SWC)*

Changes in SWC at 15 cm (SWC-15) and 30 cm (SWC-30), measured by the sensors installed in the plantation with the longest record, are shown in Figure 2. The SWC differences between SWC-15 and SWC-30 were not distinct before day of year (DOY) 180. However, after DOY 180, slightly higher values (~0.02) at SWC-30 were noticeable as compared with at SWC-15, except during the period when there was a lot of rainfall (DOY 208–225). No apparent SWC differences between SWC-15 and SWC-30 were observed at the other plantations where measurements lasted for only three months (data not shown).

From a soil health diagnosis project on tea plantations conducted concurrently in this region, we determined that tea farmers tended to start irrigation when the soil surface looked dry, although there was still plenty of water available in the subsoil. Therefore, the nearly overlapped curves of SWC-15 and SWC-30, from DOY 70 to 120 (a period received very little rainfall), may be the result of frequent irrigation during the period. Frequent irrigation tends to induce roots to remain at the surface soil layer, which can be detrimental to tea trees when environmental stresses occur, such as drought, flood, and cold. Excess irrigation also impedes crop growth due to a lack of sufficient air for root respiration [43]. We introduced this concept to the owner of one plantation around the middle of the year. He followed our suggestion to decrease the irrigation frequency afterward which resulted in the apparent separation between SWC-15 and SWC-30 starting from DOY 180. During the conversation with this plantation owner, he indicated that farmers usually do not have the instrumentation and expertise to determine the right time to start irrigation, which triggered the development of this irrigation advisory service.

The SWC values retrieved from S2 images (SWC-S2) were much smaller than those at SWC-15 and SWC-30 (Figure 2). However, positive correlations were observed between SWC-S2 and SWC-15 and SWC-30, respectively (Table 4). Sadeghi et al. [21] indicated that SWC-S2 correlated well with SWC measured at 5 cm depth. Therefore, SWC-S2 may represent the SWC at the soil surface, which is in accordance with the dry-like appearance observed by farmers. However, the uptake of water from soil depends on roots distributed at various depths. Therefore, the decision about when to start irrigation cannot be determined by the value of SWC-S2 alone.

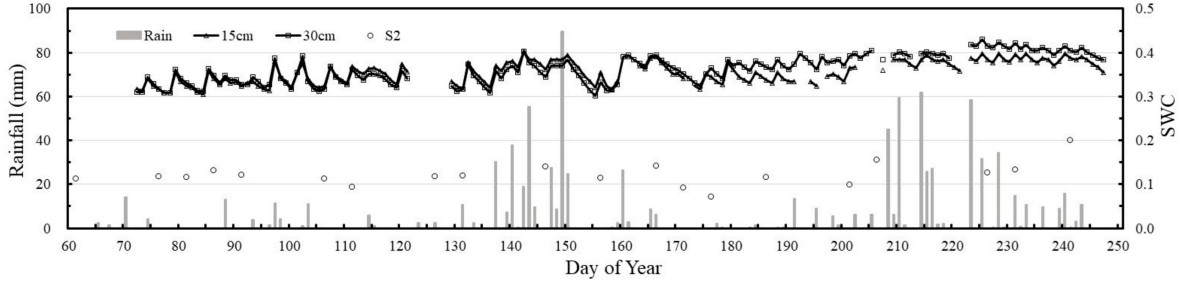

**Figure 2.** Changes in soil water content (SWC) at 15 and 30 cm depths measured by volumetric soil moisture sensors and SWC retrieved from Sentinel-2 (S2) images.

**Table 4.** Correlation coefficients among SWC retrieved from S2 images (SWC-S2), SWC measured at 15 cm (SWC-15) and 30 cm (SWC-30).

|  | **SWC-S2** | **SWC-15** | **SWC-30** |
|---|---|---|---|
| SWC-S2 | 1 |  |  |
| SWC-15 | 0.43 | 1 |  |
| SWC-30 | 0.33 | 0.90 | 1 |

### 3.2. Thresholds to Classify SWC and NDWI

The index for canopy water content (e.g., NDWI) can be used to quantify root zone soil moisture [44]. However, similar to the NDVI, the NDWI can also be affected by background interferences. Until more thorough studies have been conducted, the decision about when to start irrigation should not be determined by the value of NDWI alone. Therefore, we used the SWC and NDWI values jointly to determine the extent of water availability at a particular plantation, i.e., the CWAI defined in Section 2.3.

As shown in Table 1, level thresholds for the SWC and NDWI values retrieved from S2 images are required for determining the value of the CWAI. Instead of spending lots of time and effort running complex experiments to determine the proper thresholds for classifying the values of the SWC and NDWI into different levels, we resolved this issue by using the

distribution of the SWC and NDWI values from historical S2 images of plantations, which are known to have been well taken care of. The core concept was built upon the following assumptions:

(1)     All plantations would not irrigate at the same frequency (i.e., time and amount of irrigation water supplied are different among plantations).
(2)     Spatial heterogeneity of soil properties exists (i.e., there are internal variations within a plantation and external variations among plantations).
(3)     Plant available water in root zones differed between wet and dry seasons.
(4)     None of the plantation owners would intentionally let tea trees experience severe drought stress.

Therefore, the expectation was that large data pairs of SWC and NDWI values in wide ranges could be produced by using many plantations, from which the thresholds for level classification could be determined statistically.

Using the 121 collaborating plantations of the soil health diagnosis project conducted concurrently in the study region as target plantations and available S2 images from 2018 to June 2020, a total of 7620 plantation-averaged data pairs were produced. Although the values of SWC and NDWI are positively related (r = 0.8), as expected (Figure 3A), the existence of wide variations indicates that the value of CWAI determined by these two variables jointly is more trustworthy than by any of them individually. The accumulated frequency distributions of SWC and NDWI values are shown in Figure 3B,C. The values of SWC and NDWI at the 25th, 50th, and 75th percentiles were 0.08, 0.11, 0.15 and 0.11, 0.17, 0.22, respectively. These values were used as thresholds for the level classification in Table 1. Therefore, CWAI ≥ 4 means at least one of the indices is at a level above the 50th percentile, without the other at the level below the 25th percentile; CWAI < 3 means at least one of the indices is at a level below the 25th percentile, without the other at levels above the 50th percentile.

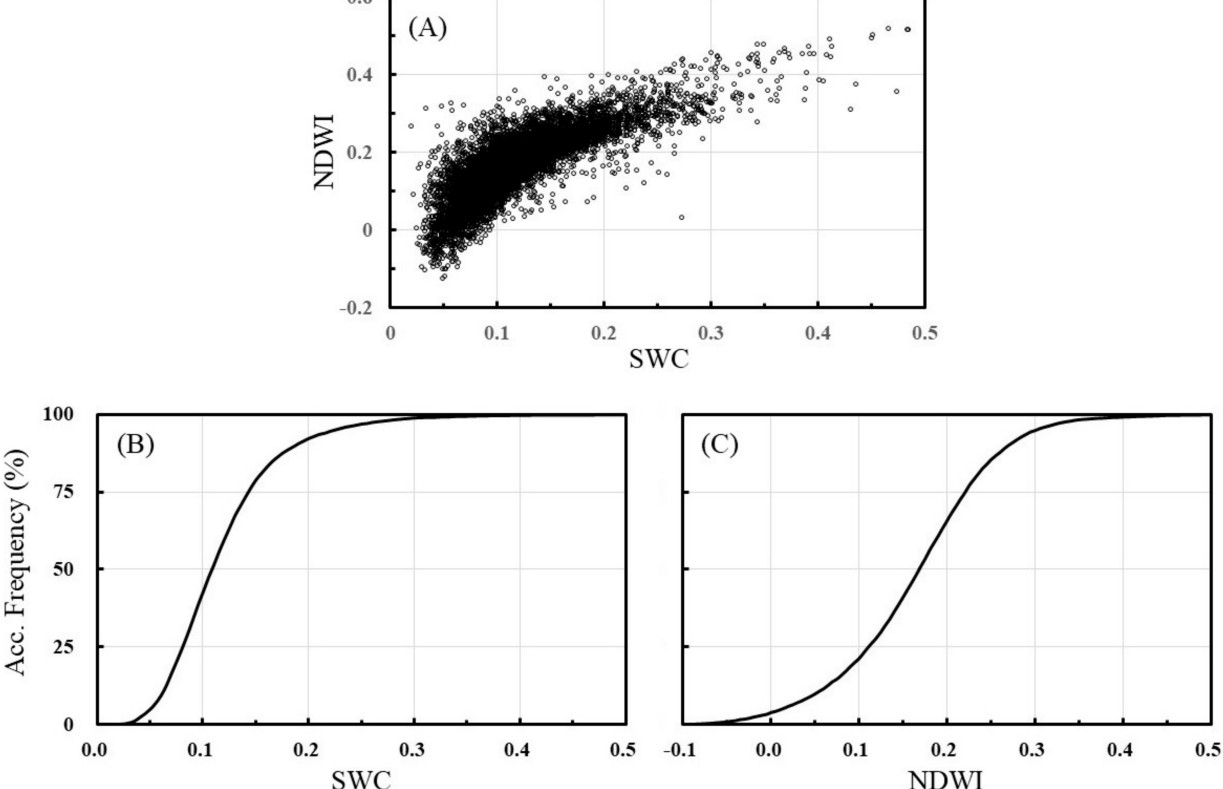

**Figure 3.** Relationships between SWC and NDWI (**A**) and accumulated frequency distributions of SWC (**B**) and NDWI (**C**).

### 3.3. Temporal Variations of SWC, NDWI, CWAI, and MSAVI at Ground Data Collecting Sites

Temporal variations of SWC, NDWI, CWAI, and MSAVI values at ground data collecting sites from 2018 to 2020 are shown in Figure 4. As discussed above, frequent irrigation has resulted in high SWC values during the dry periods (Figure 4A). The SWC values at these plantations were rarely dropped to the lowest level (<0.08), which indicated that all three farmers tended to irrigate often. In addition, the SWC values of the plantation managed by Farmer 2 (i.e., F2) were noticeably higher than the plantations managed by Farmer 1 (i.e., F1) and Farmer 3 (i.e., F3) during the dry periods and were at the highest level ($\geq 0.15$) most of the time. This indicated that Farmer 2 tended to use more irrigation water than the other two farmers. This speculation is supported by the amount of irrigation water used per unit area, converted from annual irrigation charges (Table 5). The irrigation rates at F2 exceeded those of F1 and F3. Therefore, the NDWI values at F2 were higher than that at F1 and F3 during the dry season and were at the highest level ($\geq 0.22$) most of the time because more irrigation water was supplied (Figure 4B).

**Table 5.** Amount of irrigation water used per year (t ha$^{-1}$).

| Plantation | 2017 | 2018 | 2019 | Average |
|:---:|:---:|:---:|:---:|:---:|
| F1 | 2090 | 2239 | 2429 | 2252 |
| F2 | 2876 | 2929 | 2983 | 2930 |
| F3 | 1494 | 1597 | 1749 | 1614 |

Also shown in Figure 4B, the NDWI values dropped to the lowest level (<0.11) at F3 during the period from January to May 2018, and at F1 and F2 from February to March 2019. The drop of F3 in 2018 was due to deep trimming of branches, as evidenced by the very low MSAVI values at the time (Figure 4D). The abnormally warm January and February in 2019 increased transpiration from plants, which caused the drop in the NDWI values at F1 and F2 and produced many wilted twigs (as evidenced by the drop of MSAVI values at the time). The F3 increased the irrigation at the time, as evidenced by having the highest SWC during the period, and thus maintaining the NDWI values at higher levels. This explains the apparent jump in SWC and NDWI values at all three plantations from October 2019 to January 2020, because farmers tried to avoid another severe drought stress.

As explained above, CWAI values dropped to near one at F3 in early 2018 and F1 and F2 in early 2019 (Figure 4C). Other than these two time periods, the values of CWAI rarely dropped to below three at these three well-watered plantations, which indicates that CWAI < 3 is a suitable threshold to start irrigation.

Adequate water and temperature during the summer increase the flourishing growth of tea plants, increasing the MSAVI values to a maximum (Figure 4D). Minimums of MSAVI generally appear in January and February. However, as can be observed in the figure, the values of MSAVI should be at least 0.4 for a well-maintained healthy tea plantation. This threshold is now being used to diagnose soil problems in the tea plantation health management project.

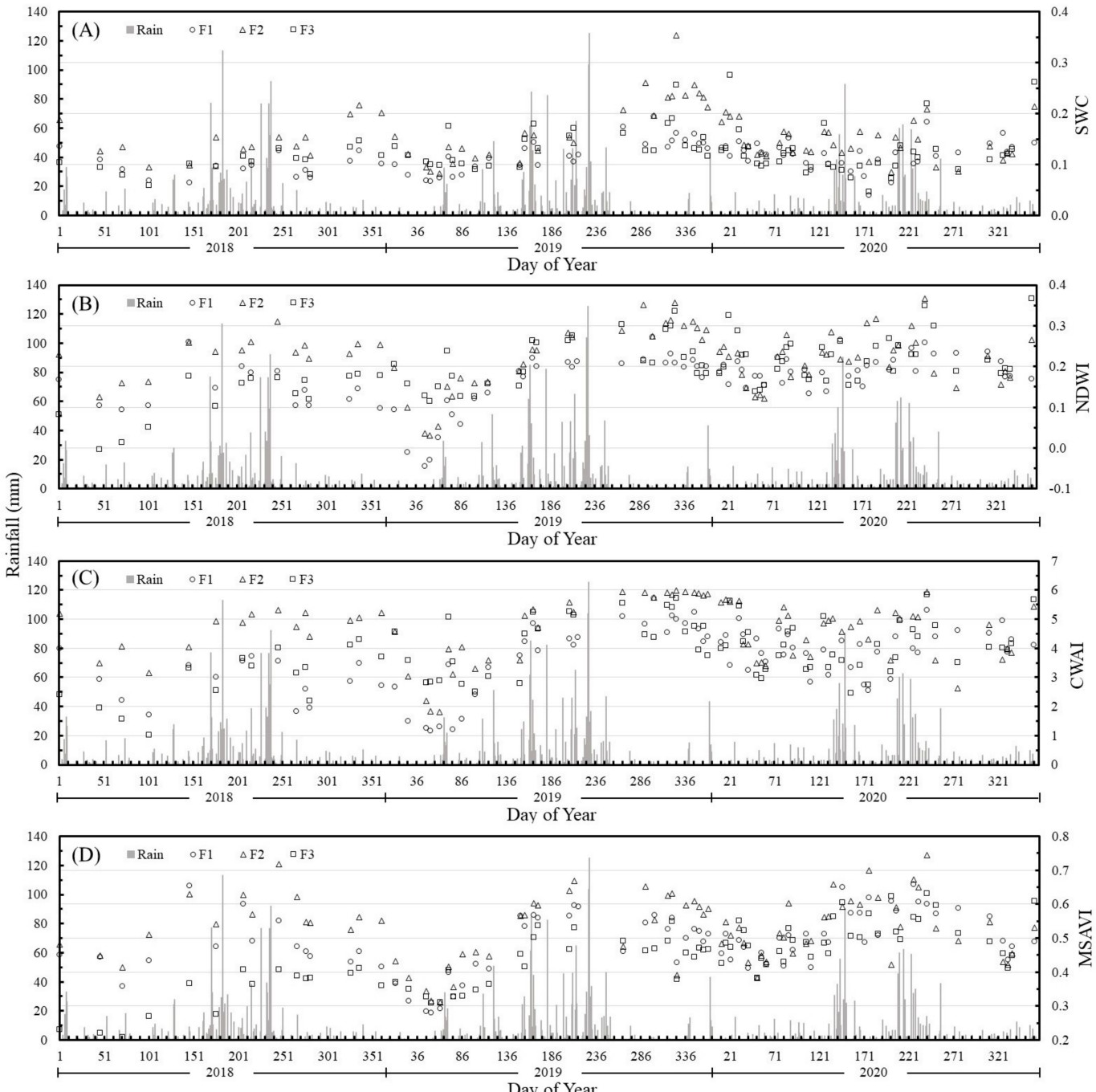

**Figure 4.** Changes in (**A**) soil water content (SWC); (**B**) normalized difference water index (NDWI); (**C**) crop water availability index (CWAI); (**D**) modified soil-adjusted vegetation index (MSAVI), at three ground data collecting plantations from 2018 to 2020.

*3.4. Spatiotemporal Variations of SWC, NDWI, CWAI, and MSAVI at Ground Data Collecting Sites*

Spatiotemporal maps of SWC, NDWI, CWAI, and MSAVI for F1, F2, and F3 from September to December 2020 and corresponding aerial images are shown in Figure 5. Non-uniform distributions of soil water status and biomass within each plantation are very apparent. For F1, areas along the southwest (SW) edge of the plantation dried out (lower SWC and NDWI values) faster than the other areas of the plantation, reflected in the CWAI values, and caused lower biomass as shown by the MSAVI values. For F2 and F3, areas along the northeast (NE) edge and near the northwest (NW) corner of the plantations,

respectively, dried out faster than the other areas of the plantations. Values of CWAI and MSAVI also responded accordingly in these two plantations.

As shown in the aerial image, the row direction of tea trees was in the NW-SE direction. Therefore, water pipelines for the sprinkler system were arranged in the NW-SE direction. When applying irrigation water, farmers manually turn on and off the pipelines in groups, due to limited incoming water controlled by the pipe diameter delivering water to the plantation, in order to maintain adequate water pressure along the pipes. The irrigation water dispensing uniformity tests (data not shown) indicated that the observed spatial variations of soil water status at F1 and F2 were mainly caused by the amount of water delivered through different pipe groups. The drier parts of F1 and F2 received less water during the same irrigation period. However, the observed soil water status variations at F3 were mainly due to differences in the types of sprinkler heads installed. The sprinkler heads installed in the NW corner area were less efficient than those installed in the other areas of the plantation. The relatively diffuse zonal boundaries of soil water status at each plantation indicated that soil properties' heterogeneity should also play an essential role in the observed spatial variations. The information derived through analyzing the spatiotemporal maps can be useful to farmers for adjusting the duration of irrigation for different pipe groups and the type and density of sprinkler heads installed. This information also allowed researchers to investigate relationships between the spatial variations of plant growth and soil water status.

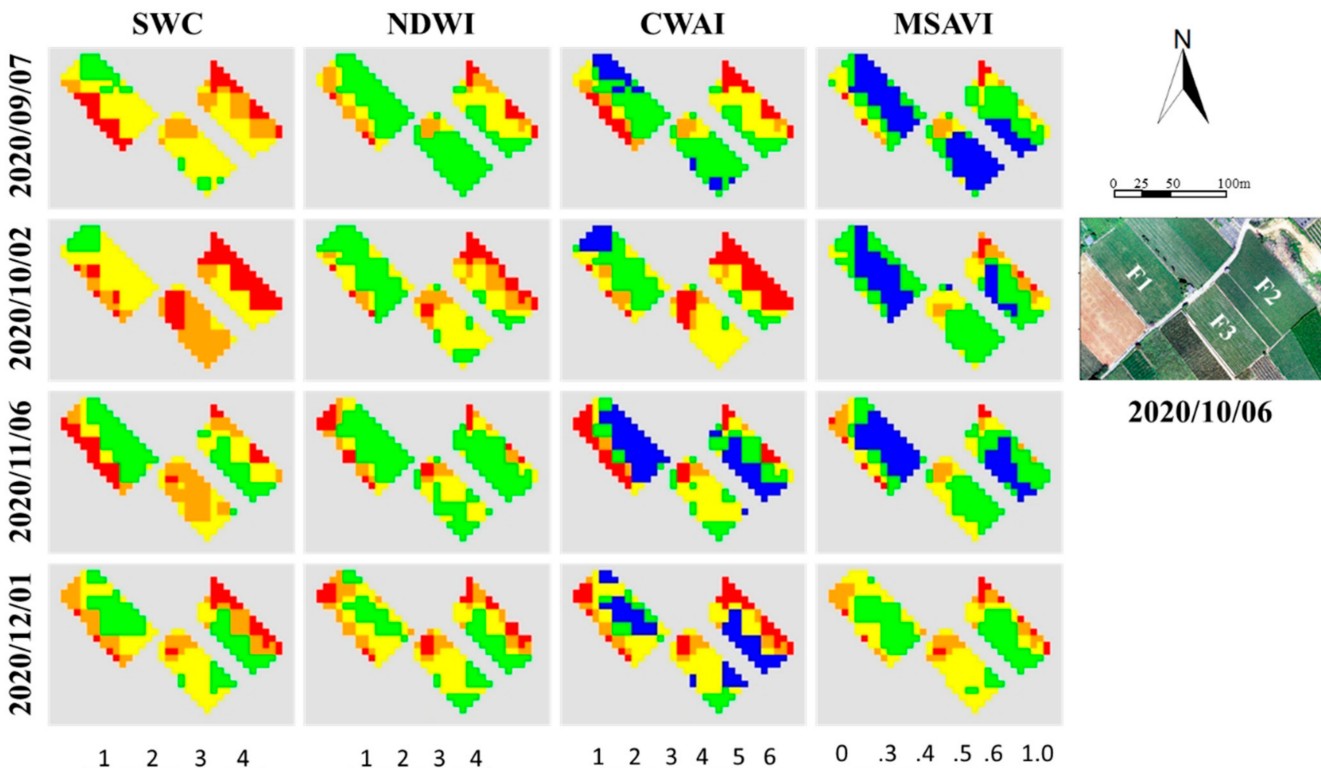

**Figure 5.** Spatiotemporal maps of SWC, NDWI, CWAI, and MSAV for three ground data collecting plantations (F1, F2, and F3) from September to December 2020 and corresponding aerial image. Scales of SWC and NDVI correspond to the four levels classified by the thresholds specified in Table 1 (1, the driest and 4, the wettest). The aerial image on 6 October 2020 was taken by a Phantom 4 Pro (DJI Science and Technology Co., Ltd., Shenzhen, China) and the orthomosaic was generated by Pix4Dmapper (Pix4D Inc., San Francisco, CA, USA).

### 3.5. Recommended Irrigation Rate per Irrigation Event

In the current version of the service, the amount of irrigation water required for a site-specific plantation was not estimated because of the complexities of estimating available

water that remained in soils. Although the reference evapotranspiration (ETo) can be estimated through images observed by meteorological satellites (e.g., [41,42]), empirical relations between crop coefficient (Kc) and NDVI (or other more appropriate vegetation indices) remain to be determined. In addition, other than precipitation records, precise information regarding soil properties and the amount of irrigation water previously applied are also required in order to run a water flow simulation model (e.g., Hydrus-1D, [45]) to estimate the amount of water that remained in the soil and the amount of irrigation water required for each target plantation. However, local irrigation specialists have indicated that an irrigation rate of 300 t ha$^{-1}$ could satisfy most upland crop needs and had the benefit of inducing downward development of root systems [46]. Therefore, an irrigation rate of 300 t ha$^{-1}$ per irrigation event is recommended in the current version.

Assuming the three previously mentioned farmers conducted irrigation practices only during the dry season (from October to March) and at an interval of 7 days, we estimate that there were about 25 irrigation events per year. According to the data provided in Table 4, this corresponds to about 60 to 120 t ha$^{-1}$ per irrigation event, which were already well below the recommended rate (300 t ha$^{-1}$). Considering that there should be more irrigation events during the dry season (an interval of 3–5 days is a common practice adopted by local farmers), the amount of irrigation water applied per irrigation event should be even less. The frequent but insufficient rate of irrigation practices currently used by local farmers have created constantly moist to wet surface soils, together with the practice of not blending applied fertilizers into the soil, may be the main factors contributing to the shallow root systems of tea trees commonly observed in the region.

### 3.6. Importance of the Service

Although the average irrigation cost represents only a small part (14%) of the total production cost, it is estimated that the duration between irrigations may extend from the common practice of 3–5 days to at least 10 days or longer, which would be greater than or equal to a 50% reduction in irrigation costs. Additionally, the saved water can help to sustain tea trees (and other crops), and therefore overcome severe drought stress that may occur. Considering the severe drought that Taiwan is currently facing as an example, water reserves of major reservoirs in the region reached an historical low in April 2021 and are still dropping [47]. Many tea trees in the studying region are not producing new leaves and gradually wilting due to the lack of irrigation water. This service, if implemented earlier and more widely, can make significant contributions to ensure rational use of the increasing water scarcity and competition for water resources.

### 4. Conclusions

In this paper, we describe the development of an inexpensive irrigation advisory service for site-specific tea plantations, which can help farmers to make irrigation decisions. The service's recommendation is based on a two-tiered approach using SWC and the NDWI values derived from free and open accessible S2 images and precipitation records published on the web by the local weather agency. The output of the service is not limited to making decisions regarding when to start, or whether to start, irrigation. Maps of crop available water status for a plantation can also be produced to improve the uniformity of water delivered by the sprinkler irrigation system. However, due to the complexities involved in estimating soil water reserve, the suggestion of a site-specific required amount of irrigation water is still not available using the current version of the service. Instead, a single irrigation rate (i.e., 300 t ha$^{-1}$) per irrigation event is suggested. This service is expected to be a useful tool for assisting farmers to make better decisions regarding the timing to start irrigation and to reduce expenses associated with buying irrigation water. Indirect benefits may include water conservation by increasing water use efficiency and deeper root systems for better tolerance of environmental stresses.

The accuracy of an irrigation advisory service relies on the availability of site-specific data and implementing a reliable decision method. The surface soil moisture content and

canopy water content of tea trees retrieved from satellite images are valuable because they provide more spatially extensive and real-time crop performance data and make up for the defects of immobile soil moisture sensors. The use of two vegetation indices, SWC and the NDWI, and statistically derived level classification thresholds reduces the chances of misclassification of alert levels. To reduce the effects of cloud obscury on the availability of S2 images, three consecutive S2 images and past precipitation records are used as auxiliary information to help determine or modify the warning level to be sent.

However, it is the farmers themselves who decide when to start the irrigation. Although several collaborating farmers indicated that they had decreased the irrigation frequency because of receiving the recommendations sent to them, it was not easy to convince them to operate as suggested because they worried about production loss. Considering that this service has already been automated and the required S2 images and precipitation records are free, we have decided to continue this service to all interested farmers until the end of 2021 to collect more local evidence and successful cases. We believe this proactive approach might improve the performance of the developed advisory service by incorporating more farmers' experiences oriented towards reducing production risks. Meanwhile, improvements to the current version, such as site-specific irrigation rate recommendations and better classification levels and thresholds employed in the two-tiered approach, are also planned.

**Author Contributions:** Conceptualization, methodology and software Y.S.; investigation, data curation, validation, and visualization Y.-P.W.; writing—original draft preparation, Y.-P.W. and Y.S.; writing—review and editing, C.-T.C. and Y.-C.T.; funding acquisition, Y.S. All authors have read and agreed to the published version of the manuscript.

**Funding:** This research was financially supported by the "Innovation and Development Center of Sustainable Agriculture" from the Featured Areas Research Center Program within the framework of the Higher Education Sprout Project by the Ministry of Education, and the Ministry of Science and Technology, Taiwan, ROC (grant numbers MOST 107-2321-B-005-016, MOST 108-2321-B-005-010, and MOST 109-2321-B-005-020).

**Institutional Review Board Statement:** Not applicable.

**Informed Consent Statement:** Not applicable.

**Data Availability Statement:** The data presented in this study are available on request from the corresponding author. The data are not publicly available due to privacy concerns.

**Acknowledgments:** Sentinel-2 images provided by ESA and field data collected by Te-Chou Liao and Pei-Chen Tang are greatly appreciated.

**Conflicts of Interest:** The authors declare that they have no known competing financial interest or personal relationships that could have appeared to influence the work reported in this paper.

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
