# Peer review of "A Sentinel-2 Image-Based Irrigation Advisory Service: Cases for Tea Plantations"

_water, doi:10.3390/w13091305_

Round 1

Reviewer 1 Report

Overall, the article was interesting to read. The authors created a service that may help the farmers not only from Taiwan but also from other countries to do proper irrigation of their farms, decrease the costs spent on it, and meet Sustainable Development Goal (SDG) 6.4 of the United Nations. The idea is great, and it should be supported by all countries in the world, especially when instruments to realize it (Sentinel satellite systems) are already available and in use.

The authors demonstrated a good connection between science and agriculture, which is missed nowadays in many scientific institutions.

The article consists of a decent study, but the results have still critical questions for further publishing.

Line 55: 14% is already a small part of the total costs. Could you, please, estimate the reduction in total costs after using your service? It would be great to put such estimation in the Results and Discussions section.

Line 65: Please, cite more recent publications (in the case of [7], for example).

Line 66-67: I guess you wanted to say “However, the approach using only soil moisture sensors (tensiometers)…”. Here, it is important here to indicate which sensors were used. Of course, local measurements of soil moisture content can have a fairly large error. The entire paragraph about soil moisture sensors weakly reveals the use of sensors, and only one citation to an article in which sensors didn't show the desired result is not suitable for comparison. I highly recommend to give more examples of the use of sensors in order to identify their advantages and disadvantages.

Line 133-134:  23.80144-23.89009 N, 120.61529−120.73726 E

Line 150: From 1.5 m to 0.3 m? I think the second number is wrong.

Line 192: What was the reason to select NDWI, and not MSAVI, for example? Could you, please, explain it in the article?

Line 199: Please, show the Eq. for SWC.

Line 200: Use the standard reference style for this journal.

Line 200-201: Repeat the references for each index.

Line 271: It is a pity that no sensors were installed at <5cm depth to verify SWC-S2.

Line 285-288: How was other information (except rainfall) taken into account in your study?

Line 305: Support this statement with a reference.

Line 312: Explain, please, the large difference between SWC-S2 and SWC-15/SWC-20 especially before DOY 180 when SWC-15 was very close to SWC-20 (irrigation was frequently done; Line 302). It seems to be that the difference should be smaller than those after DOY 180 when irrigation was not frequent (Line 307). But the difference between SWC-S2 and SWC-15/SWC-20 is nearly the same during the whole period of measurements.

Line 314: These data are quite important to show correlation S2-SWC with onsite measurements by the sensors. I suggest to show such a correlation on a plot.

Line 325: Support this statement with a reference.

Line 348. Could you please explain the reason why each parameter, SWC (Line 319) and NDWI (Line 328), individually is not suitable for the decision making, but their weighted sum (Table 1) is good for this, moreover the correlation between SWC and NDWI is quite high (Line 348).

Line 397: Table 1 has only integer values of CWAI. Most likely, Figure 3C has an average CWAI over the field or something else. Please, include such an explanation in the text of the article.

Reviewer 2 Report

Lines 66 to 69: Is the soil variability a concern for small-scale farming (or targeted farmers in this study)? It would be great to add a satellite image showing the studied Tea farms, and on the side, you can add the soil mal to show the variation. This will help your readers get familiar with the study area and follow along with the discussions. 

Line 99: "PROSPECT" is it an abbreviation for something or a tool you have used? Please use the full form at first mention. 

Line 104: "]at" please fix the error. 

Line 104: Please include the full form of NDWI at first mention. I see it's mentioned on the next page, but please move it here. 

Line 185: What is ESA?

Lines 222-233: Have a flow chart or process map to explain the approach would be helpful for readers to follow along with the explanation. 

Line 315: "(2017)" please fix the citation style to match with the rest of the paper. 

Line 412: "As shown in the aerial image..." Please move the Figure to this page. 

Line 426: Please include the most significant maps and explain how farmers would interpret the irrigation duration from the maps? You have mentioned that farmers are notified by text message to start or stop the irrigation. But here you say that the "map would be useful to farmers". Please explain how. 

Lines 440-443: Is this recommendation purely based on the instincts of the local irrigation specialists? Are they farmers or agricultural scientists? What is the size of the panel, experience level, etc.?

Figure 4: What is the significance of the scale? They need to be normalized to compare with different parameters. 

Round 2

Reviewer 1 Report

Figure 2b, c: Y-axis grid lines (0, 25, 50, 75, 100) instead of (0, 20, 40, 60, 80, 100) would better fit the text.

Figure 5. Please, specify the source of the aerial image.

Ref. 37 has the wrong title.

For the future, I would suggest the following improvements to the service (probably, a new study):

- Figure 3a: try to get rid of the correlation between NDWI and SWC (something like NDWI/f(SWC)), where f(SWC) means some function of SWC. In this case, I think, your Table 1 will more accurately describe the condition of the tea plantations;

- collect the outcome (yield) of the 121 collaborating plantations (Line 358) and take it into account for determining the threshold levels. 

Round 3

Reviewer 1 Report

Thank you, and I hope that your service will become popular in Taiwan.